# Storytelling of Myocarditis

**DOI:** 10.3390/biomedicines12040832

**Published:** 2024-04-09

**Authors:** Gaetano Thiene

**Affiliations:** Department of Cardiac, Thoracic, Vascular Sciences, and Public Health, University of Padua, 35121 Padua, Italy; gaetano.thiene@unipd.it

**Keywords:** ECMO, endomyocardial biopsy, history of medicine, molecular pathology, myocarditis

## Abstract

In 1900, Fiedler first reported autopsy cases with peculiar inflammation of the myocardium, which he named interstitial myocarditis. He postulated an isolated cardiac inflammation of the myocardium in the absence of multiorgan involvement and with a poor prognosis due to invisible microorganisms, which years later would have been identified as viruses. The revision of original histologic sections by Schmorl showed cases with lymphocytes and others with giant-cell inflammatory histotypes. The in vivo diagnosis of myocarditis became possible thanks to right cardiac catheterization with endomyocardial biopsy (EMB). The gold standard for diagnosis was achieved with the employment of immunohistochemistry and molecular investigation by Polymerase Chain Reaction (PCR), which allows for the detection of viruses as causal agents. Both RNA and DNA were revealed to be cardiotropic, with a common receptor (CAR). A protease, coded by coxsackie virus, disrupts the cytoskeleton and accounts for cell death. Unfortunately, vaccination, despite having been revealed to be effective in animal experiments, has not yet entered the clinical field for prevention. Cardiac Magnetic Resonance turned out to be a revolutionary tool for in vivo diagnosis through the detection of edema (inflammatory exudate). Myocarditis may be fulminant in terms of clinical presentation but not necessarily fatal. The application of ExtraCorporeal Membrane Oxygenation (ECMO) allows for relieving the overloaded native heart.

## 1. Introduction

The history of non-ischemic inflammation of the heart muscle dates back when the study of morbid anatomy moved from a gross view of organs by the naked eye (Figure 1) to cell pathology studied through a microscope.

Robert Hooke (1635–1703) (Figure 2), in 1665, invented the microscope, using magnifying glasses to see minute bodies within the organs [1]. He observed thin slices of cork identifying empty spaces, which he called “cells” because they were similar to the cells of monks. The title of his book was quite meaningful (Figure 2).

The use of the microscope became routine in Germany in the 19th century with the Berlin School, which had supremacy in pathology. The discovery of cells as elementary units of organs was a fundamental step forward compared to the methods of Giovanni Battista Morgagni (1682–1771) and Karl von Rokitansky (1804–1878) (both organ pathologists), and the title of Rudolf Virchow’s (1821–1902) book was a paradigm (Figure 3). Within cellular pathology, with the origin of each cell coming from another cell, Virchow put an end to the Hippocratic and Galenic theories of the humors.

The microscope allowed for the discovery of microorganisms (bacteria) (called “seminaria” by Fracastoro)—the cause of contagious diseases. This was similar to the efforts of Robert Koch (1843–1910), who discovered tuberculosis (Figure 4).

In 1900, Carl Ludwig Alfred Fiedler (1835–1921) (Figure 5) reported an acute heart failure in young people who died suddenly with fulminant, malaria-like fever in the absence of coronary, valve, pericardial disease, and he called it interstitial myocarditis [2]. Through a microscope, interstitial inflammatory infiltrates of the myocyte were observed, which spared other organs (isolated myocarditis). He believed it was the consequence of an atypical infection like scarlet, diphtheria, or typhus, different from acute rheumatic fever or syphilis. The prognosis was severe and lethal within 5–17 days. Oddly enough, the visionary Fiedler advanced the possibility of recovery of cardiac contractility because, despite the severe inflammatory phenomenon, the myocardial injury appeared potentially reversible because of the absence of significant cardiomyocyte death.

An autopsy case of similar myocardial inflammation had been published three years earlier by Abramov [3].

In 1905, Saltykow questioned the distinction between parenchymatous (diphtheric) and interstitial myocarditis and dictated the definition of myocarditis as a non-ischemic disease of the myocardium with inflammatory infiltrates [4].

A revision of Fiedler’s original histologic slides, performed by Christian Georg Schmorl (1861–1932) (Figure 6), showed either a lymphocytic or giant-cell inflammatory histotype, which years later would be interpreted as viral or immune myocarditis. Since then, the latter has been labeled with the eponym “Giant Cell Myocarditis”.

Another type of myocarditis was rheumatic “pancarditis”, first observed by Karl Albert Ludwig Aschoff (1866–1942) [5] (Figure 7). It was not an infective disease, since the culture of valve vegetations (“verrucae”) turned out to be sterile. Granulomatous myocardial inflammation was observed by Aschoff with “bodies” consisting of owl’s eye cells (Aschoff cells) and caterpillar cells (Anitschkow cells) (Figure 7). The concept of an auto-immune disease, triggered by a streptococcus pharyngeal infection, was proven later.

On the other side of the ocean, Carlos Chagas (1879–1934), a Brazilian physician, reported severe myocarditis in the setting of a systemic infective disease by the protozoan Trypanosoma cruzi [6] (Figure 8). Despite working in primitive conditions, Chagas described a previously unknown infectious disease in detail (pathogen, vector, host, clinical manifestations, epidemiology, and demographics) and observed the infective organism in the inflamed myocardium under the microscope.

In 1929, Mitchell Bernstein wrote of the heart’s involvement in sarcoidosis [7], focusing on non-caseous granulomatous giant-cell myocarditis, quite different from tuberculosis and most probably immune in nature. This typically involves the right ventricle and triggers ventricular arrhythmias, mimicking arrhythmogenic cardiomyopathy (Figure 9). Unlike the latter, it may also involve the ventricular septum and is easily and pathognomonically diagnosed in vivo through right-ventricular endomyocardial biopsy.

Myocarditis has various etiologies and mechanisms (infectious, immune, allergic, or still idiopathic). Viral infection is the main cause of lymphocytic myocarditis in the Western world.

For years, the diagnosis of viral myocarditis had been based on serology, by revealing high levels of antiviral antibodies, or by culturing blood or pericardial effusion.

In 1948 a small RNA enterovirus was identified as cardiotropic by Gilbert Dallford (1900–1979) (Figure 10) at the Department of Health, Albany, within the feces of children in a village named Coxsackie, New York State [8].

Memorable reviews were accomplished by Otto Saphir in 1941 on 240 cases of postmortem myocarditis [9] and by Jack Woodruff in 1980 on animal experiments of viral myocarditis, with particular emphasis on coxsackie B enterovirus [10].

As far as etiology is concerned, myocarditis can be due to infective microorganisms (Table 1), allergens, drugs, toxic agents, and immune reactions. Signs and symptoms as phenotypical expressions of myocarditis are heart failure, cardiogenic shock, myocardial infarction-like, chest pain, arrhythmias, syncope, and sudden death (Figure 11). It may even be asymptomatic.

Table 1 reports a panel of infective agents (viruses, bacteria, mycobacteria, fungi, protozoa, rickettsiae, chlamydia, …). The microorganisms may be visible inside the cardiomyocytes (Figure 12).

Parvovirus B19 is mostly a vasculotropic virus, and herpes virus infects mesenchymal cells.

Autoimmune myocarditis, occurring in systemic lupus erythematosus (SLE), does not lead to a rapid decline in contractility because the main target is conducting tissue, with the onset of sino-atrial and atrioventricular blocks.

Classifications according to the inflammatory histotype are: lymphocytic, neutrophilic, eosinophilic, granulomatous, and giant-cell myocarditis (Figure 13).

As far as therapy is concerned, immunosuppressive therapy is effective in immune myocarditis and more rarely in viral myocarditis (Figure 14).

In the original WHO classification of cardiomyopathies of 1980, myocarditis was put within secondary myocardial diseases, like amyloidosis. In the WHO’s revised classification of 1996, myocarditis was definitively put within primary cardiomyopathies [12], and this has been maintained.

Ventricular arrhythmias (VAs) are frequently associated with myocarditis. Polymorphic VAs are more common during active inflammation, while monomorphic VAs are associated with healed myocarditis [13].

The rate of myocarditis among juveniles (<35 years old) as a cause of sudden cardiac death was 14% in the prospective study of the Veneto region (Figure 15).

Arrhythmogenic triggers are the cytokines released by inflammatory cells, myocyte necrosis, interstitial edema, and fibrosis following healing [10,14]. A defibrillator jacket is recommended for a few months to prevent SD and protect patients.

The pathophysiological process of viral myocarditis may be divided into three phases (Figure 16):(1)Direct viral myocardial injury.(2)Secondary immune-mediated myocardial damage.(3)Myocardial scarring with remodeling, leading to dilated cardiomyopathy [14].

Myocardial inflammation is frequently associated with genetically determined cardiomyopathies, even at the onset of the disease.

Myocarditis has been one of the mechanisms postulated in the pathophysiology of AC since its origin [15].

Virus-negative, probably autoimmune myocarditis was found at EMB in the so-called “hot phase” of arrhythmogenic cardiomyopathy (AC), especially in children [16].

Transgenic mice overexpressing desmocollin-2 were found to develop AC associated with myocardial inflammation [17].

**Figure 16 biomedicines-12-00832-f016:**
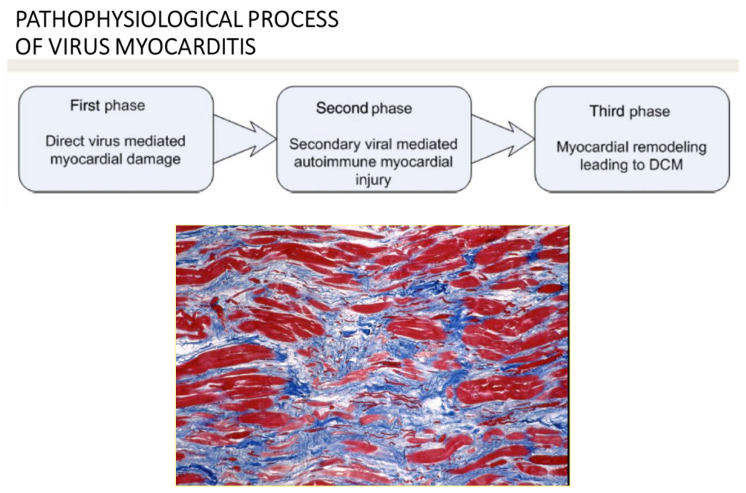
Pathophysiological process of virus myocarditis, with end-stage replacement fibrosis, resulting in dilated cardiomyopathy. Azan Mallory stain. From [18], modified.

As far as diagnosis, endomyocardial biopsy (EMB) is nowadays an essential tool. In this storytelling article, we must recall the mythical first right-ventricular catheterism by Werner Forssmann (1904–1979) [19], who, in 1929, inserted a urological catheter into his own RV ventricle through the left radial vein (Figure 17).

In 1962, the technique of transvenous EMB was introduced in Japan by Shigeru Sakakibara (1910–1979) [20] (Figure 18). At that time, investigating the myocardium was feasible only during an autopsy, since in vivo muscle biopsy was possible only at the skeletal level.

EMB via a transdermal approach can also be accomplished in the left ventricle without risk [21].

At the start, renowned scientists raised concerns about the endomyocardial biopsy procedure, both in terms of risk and sensitivity and whether the biopsy specimens could represent the state of the whole heart [22].

Conversely, EMB has been found to be effective in achieving the diagnosis of active myocarditis, since the inflammation is widespread in the myocardium with negligible error (Figure 19), if the number of samples is adequate.

Diagnostic histologic criteria were put forward in 1985 by the Society for Cardiovascular Pathology in Dallas, based upon the microscopic observation of inflammatory infiltrates associated with myocardial necrosis (“Dallas Criteria”) [23] (Figure 20). However, myocardial necrosis can be absent, despite a severe inflammatory infiltrate (Figure 21), and the Dallas Criteria have recently been “sentenced to death” [24]. Inflammatory infiltrates and interstitial edema are the diagnostic “condition sine qua non”.

Equally important is the evidence of viral cause found through Polymerase Chain Reaction (PCR), invented by Kary Mullis (1944–2019) (Figure 22) in 1986 [25], who was rewarded with the Nobel Prize in 1993.

A few years later, in 1986, N.E. Bowles et al., applying an in situ molecular hybridization technique, identified the coxsackie virus in human biopsy samples of active myocarditis and dilated cardiomyopathy [26]. EMB with histology, immunohistochemistry, and molecular analysis is nowadays the diagnostic gold standard of viral myocarditis [27].

Cardiac Magnetic Resonance (CMR) with contrast enhancement was found to be a revolutionary tool for the non-invasive diagnosis of myocarditis; it is able to detect myocardial inflammation by evidence of “edema” (fluid exudate) with a sensitivity of around 70% [28]. However, EMB remains needful to achieve the diagnosis with precise etiology and histotype for appropriate therapy with prognostic implications (for example, lymphocytic for viral etiology; granulomatous non-caseous and giant cells for the immune mechanism; and eosinophilic for the allergic mechanism). As for the outcome, viral myocarditis shows the worst prognosis [29].

Myocardial damage of myocarditis in animals was proven to be the consequence of direct cytotoxicity and/or humoral or cell-mediated immune reaction [14].

The results of a myocarditis molecular study in EMB showed a rate of viral cause of 58% in children and 71% in adults, with a prevalence of enterovirus in both (Figure 23).

Even in juvenile sudden death due to viral myocarditis, in our experience, enterovirus (coxsackie) accounted for 50% of causative viruses (Figure 24 and Figure 25).

Fulminant myocarditis in the past was considered lethal. Nowadays, with the employment of ECMO (ExtraCorporeal Membrane Oxygenation) and corticosteroid therapy, a good prognosis is feasible, despite the fact that specific antiviral drug therapy is still missing.

A ventricular assistance device using ECMO was introduced by Robert Bartlett (1939–) (Figure 26) and may temporarily supply myocardial contractility until spontaneous recovery.

The clinical presentation may be fulminant; however, spontaneous resolution may occur. The application of ECMO allows for temporary native heart rest, supplying cardiac contractile and respiratory functions. Fortunately, lymphocytic viral myocarditis is not necessarily associated with severe irreversible myocyte necrosis (Figure 21).

Since 1991, fulminant myocarditis with severe subitaneous onset and pump failure has been proven to be frequently solved [30], most probably because of less or reversible myocardial damage.

A contractile recovery with survival in patients with fulminant myocarditis and ECMO varied from 60% to 80% (Figure 27), suggesting that aggressive temporary support achieves an excellent outcome.

The coxsackievirus B genome encodes two proteases: protease 2A and protease 3C. Protease 2A cleaves dystrophin [31] (Figure 28).

Coxsackievirus B (RNA virus) and Adenovirus (DNA virus) share the same cell membrane receptor: CAR [32] (Figure 29).

Therapeutic perspectives are represented by:-Viral vaccination against coxsackievirus B3 [33], as done with the polio vaccine and the mumps/measles/rubella vaccines.-Protease inhibitors to prevent progression to chronic dilated cardiomyopathy [34].-Antiviral therapies.-Cardiac transplantation.

The latter can represent a lifesaving therapeutic option, although giant-cell immune myocarditis may relapse in the graft donor heart [35]. Cardiac transplants can be accomplished either electively in chronic heart failure or during the emergency of cardiogenic shock by acute myocarditis.

**Figure 27 biomedicines-12-00832-f027:**
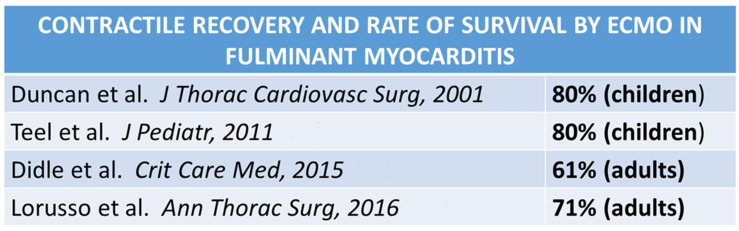
Contractile recovery and rate of survival by ECMO in fulminant myocarditis. Data from different reports [36,37,38,39].

**Figure 28 biomedicines-12-00832-f028:**
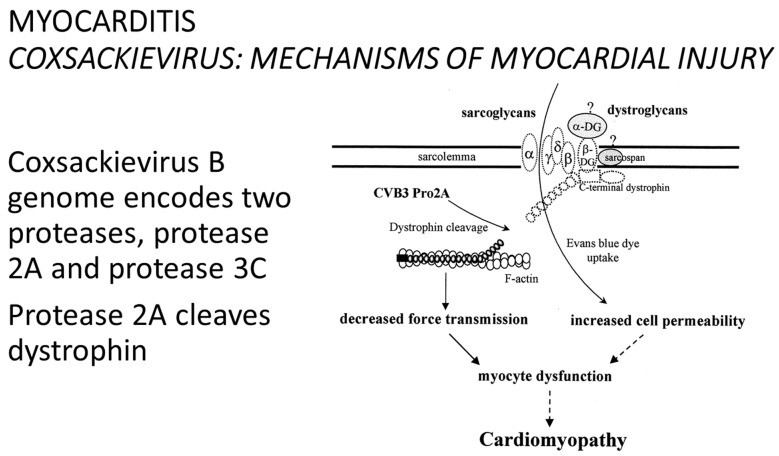
In 1999, Badorf et al. discovered protease released by coxsackievirus, cleaving dystrophin. From [40], modified.

**Figure 29 biomedicines-12-00832-f029:**
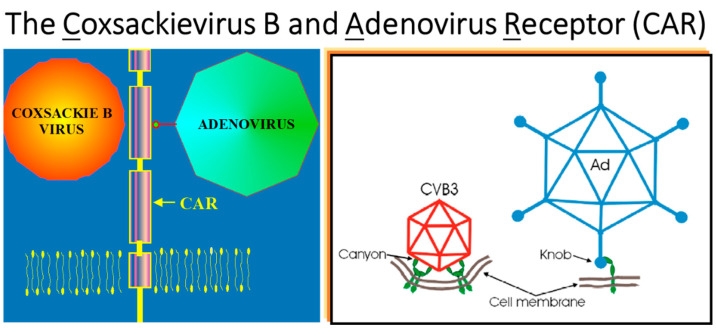
In 2001, He et al. [32] discovered that coxsackievirus and adenovirus share a common receptor (CAR).

## 2. Highlights

-Viral genomes can be detected in the myocardium of patients through endomyocardial biopsy with molecular analysis.-Edema is regularly associated with inflammation due to myocarditis and nowadays is specifically detected by Cardiac Magnetic Resonance with late enhancement.-Enterovirus infection may result in cytoskeletal lysis with cell death, entailing a severe prognosis.-Fulminant myocarditis with severe non-ischemic pump failure can heal with temporary support of myocardial contractility using ECMO, until spontaneous recovery.-The possibility of viral vaccination should be pursued.

## 3. Conclusions

Historical milestones in the field of myocarditis may be summarized as follows:
1665Hooke R. *Micrographia*, or some physiological descriptions of minute bodies provided by magnifying glasses. [1]1900Fiedler A. Acute interstitial Myocarditis in sudden heart failure. [2]1904Aschoff KAL. Acute myocarditis in the setting of rheumatic pancarditis. [3]1928Chagas C. Sur les alteration du Coeur dans la trypanosomiases américaine. [6]1929Forssmann W. Die Sondierung des rechten Herzens [Probing the right heart]. [19]1929Bernstein M et al. Sarcoid: report of a case with visceral involvement. [7]1948Dalldorf G et al. An Unidentified, Filtrable Agent Isolated From the Feces of Children. [8]1962Sakakibara S and Konno S. Endomyocardial biopsy technique. [20]1980Woodruff JF. Viral myocarditis: a review. [10]1986Mullis KB et al. Process for amplifying, detecting, and/or-cloning nucleic acid sequences. [25]1986Bowles NE et al. Detection of coxsackie-B-virus specific RNA sequences in myocardial biopsy samples from patients with myocarditis and dilated cardiomyopathy. [26]1987Aretz HT et al. Myocarditis: a histopathologic definition and classification. [23]1991Lieberman EB et al. Clinicopathologic description of fulminant myocarditis. [31]1996Richardson P et al. WHO/ISFC Cardiomyopathies Classification and Definition. [12]1999Badorff C et al. Enteroviral protease 2A cleaves dystrophin: evidence of cytoskeletal disruption in an acquired cardiomyopathy. [31]1999Kawai C. From myocarditis to cardiomyopathy: mechanisms of inflammation and cell death: learning from the past for the future. [14]2001He Y. et al. Interaction of coxsackievirus B3 with the full length coxsackievirus-adenovirus receptor. [32]2003Frustaci A et al. Immunosuppressive therapy for active lymphocytic myocarditis: virological and immunologic profile of responders versus nonresponders. [11]2009Friedereich MG et al. Cardiovascular magnetic resonance in myocarditis. [28]2013Caforio A et al. Current state of knowledge on aetiology, diagnosis, management, and therapy of myocarditis: a position statement of the European Society of Cardiology. [29]2013Thiene G et al. Diagnostic use of the endomyocardial biopsy: a consensus statement. [27]

## Figures and Tables

**Figure 1 biomedicines-12-00832-f001:**
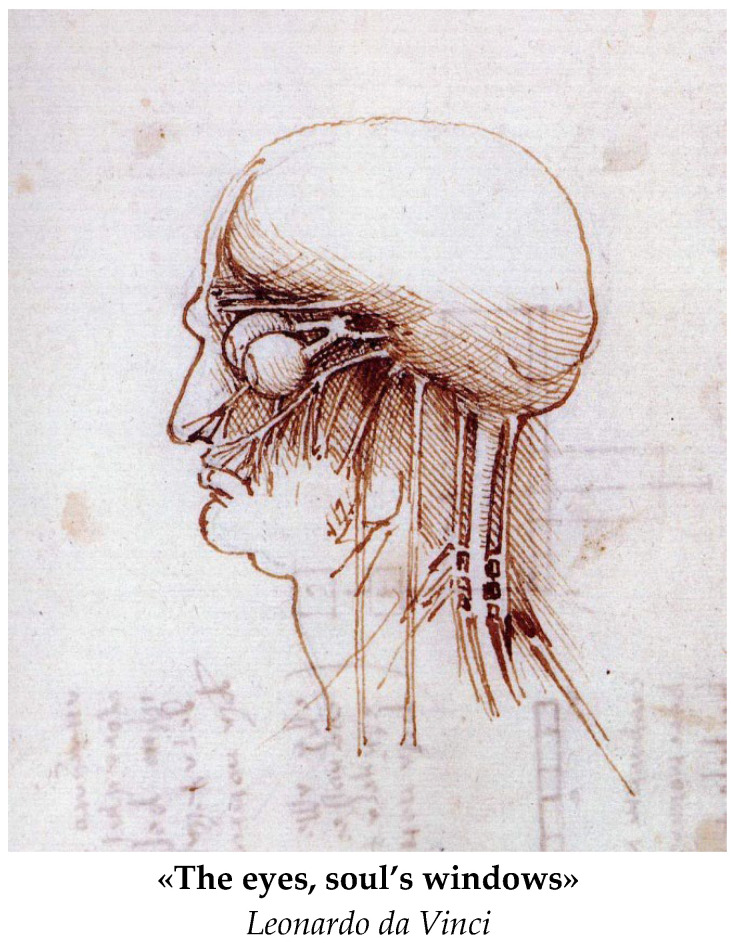
In 1510, a drawing was made by Leonardo da Vinci in his mind, for eyes are the windows of the soul with which to explore nature.

**Figure 2 biomedicines-12-00832-f002:**
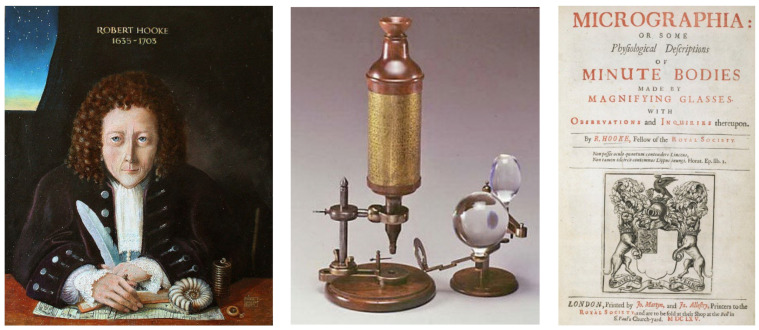
In 1665, Robert Hooke (1635–1703) invented the microscope, for the observation of “minute bodies”.

**Figure 3 biomedicines-12-00832-f003:**
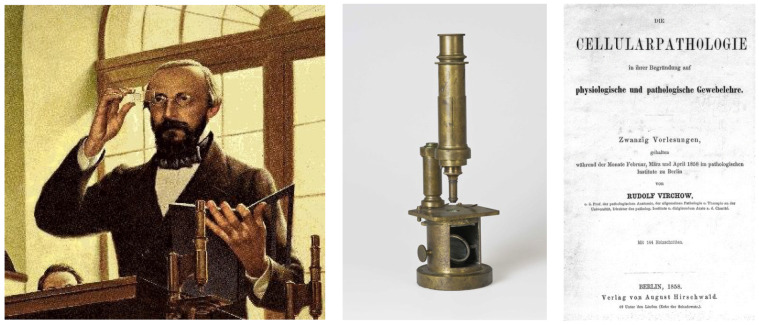
Thanks to the microscope, Rudolph Virchow (1821–1902) advanced the theory of cell pathology in the book entitled *Cellularpathologie* (Cellularpathology).

**Figure 4 biomedicines-12-00832-f004:**
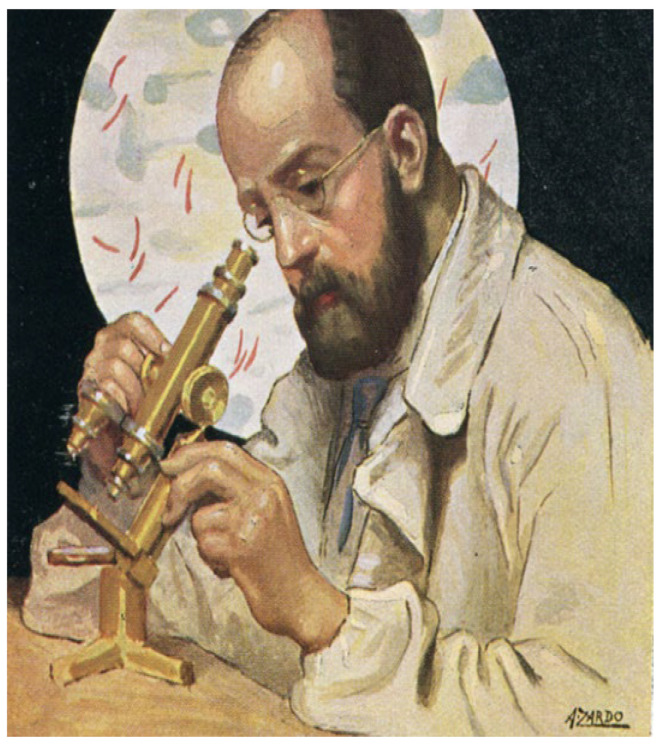
In 1882, Robert Koch (1843–1910) discovered mycobacterium tuberculosis, looking through the microscope.

**Figure 5 biomedicines-12-00832-f005:**
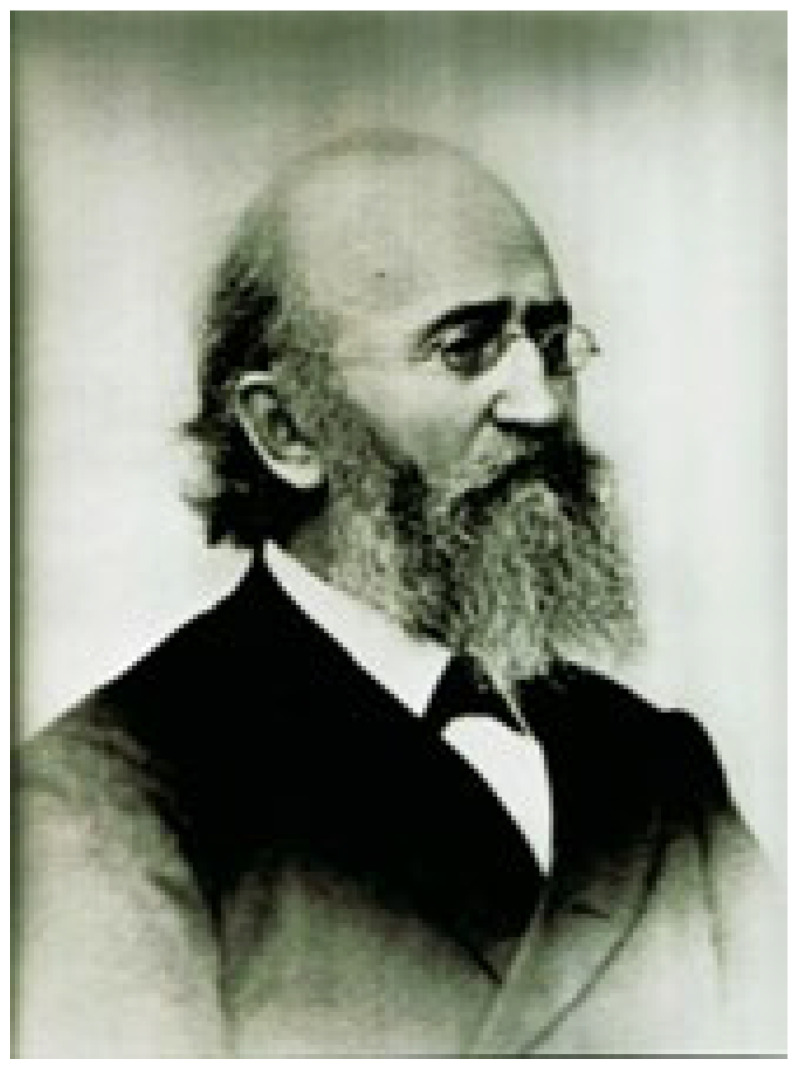
In 1900, Carl Ludwig Alfred Fiedler (1835–1921) discovered inflammatory cells within the interstice of the myocardium (interstitial myocarditis).

**Figure 6 biomedicines-12-00832-f006:**
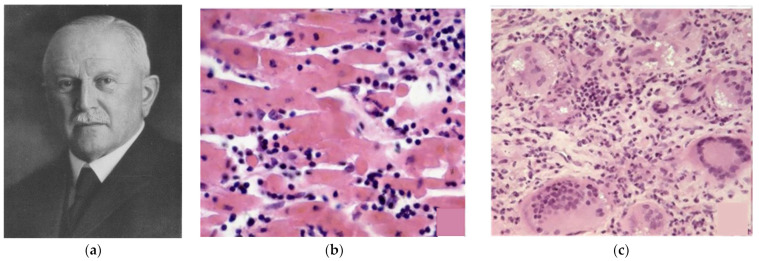
Christian Georg Schmorl (1861–1932) (**a**), by reviewing the original slides of Fiedler, observed two cellular histotypes of interstitial myocarditis: lymphocytic (**b**) and giant cells (**c**). Haematoxylin and Eosin stain.

**Figure 7 biomedicines-12-00832-f007:**
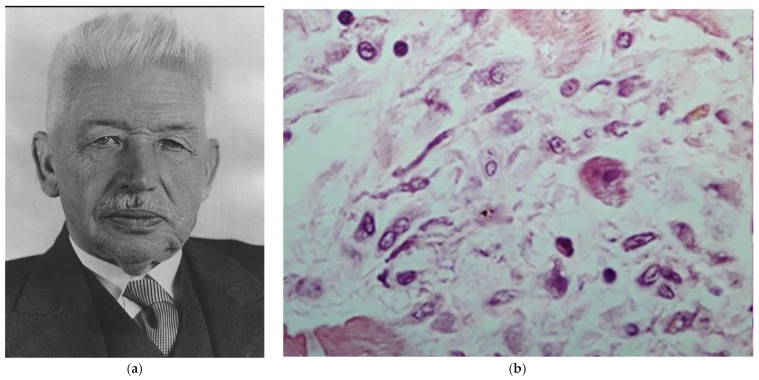
In 1900, Karl Albert Ludwig Aschoff (1866–1942) (**a**) observed myocarditis in rheumatic fever, consisting of granulomatous “bodies” with one eye and caterpillar cells (so-called rheumatic nodule) (**b**). Haematoxylin and Eosin stain.

**Figure 8 biomedicines-12-00832-f008:**
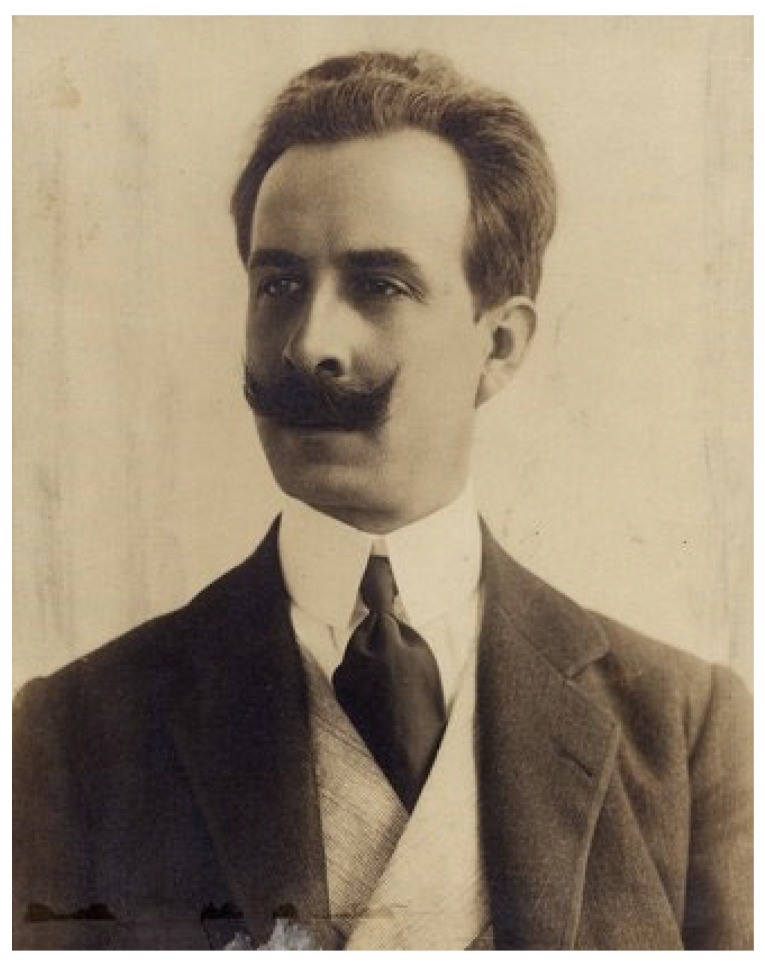
Carlos Chagas (1879–1934) first described Trypanosoma cruzi myocarditis.

**Figure 9 biomedicines-12-00832-f009:**
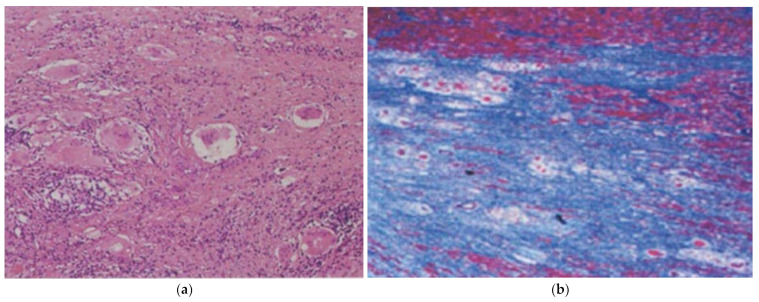
In 1929, Michel Bernstein first reported myocarditis by sarcoidosis, consisting of non-caseous granuloma with giant cells (**a**–**c**). In this specimen, both the ventricular septum and the right-ventricular free wall are involved (**d**). Haematoxylin and Eosin stain.

**Figure 10 biomedicines-12-00832-f010:**
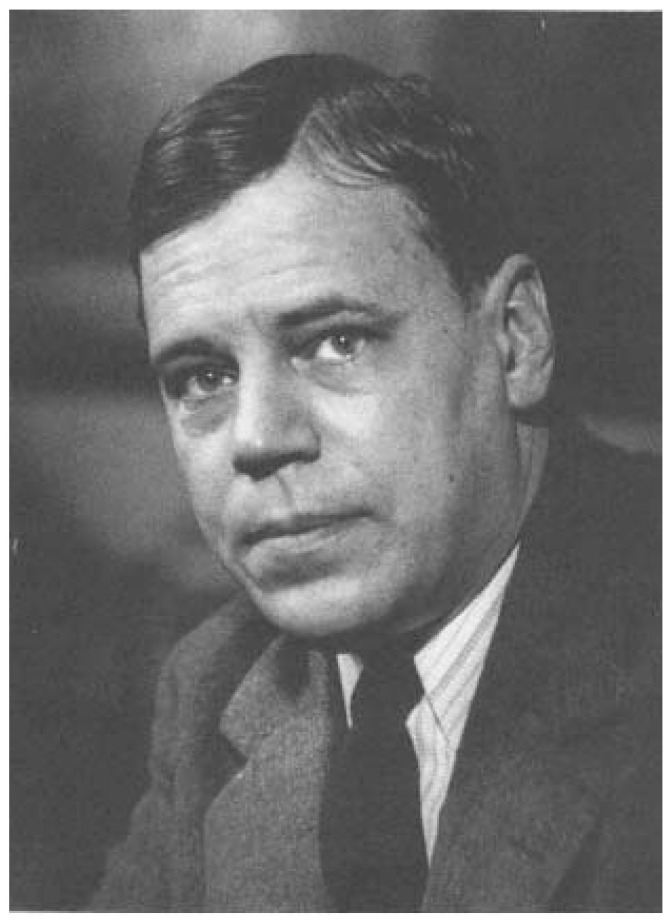
Gilbert Dallford (1900–1979), the discoverer of coxsackie virus myocarditis.

**Figure 11 biomedicines-12-00832-f011:**
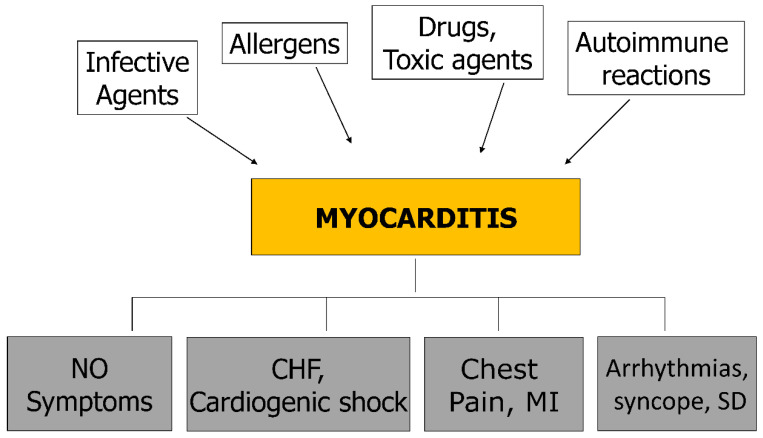
Causes, mechanisms and symptoms of myocarditis. CHF = congestive heart failure; MI = myocardial infarction; SD = sudden death.

**Figure 12 biomedicines-12-00832-f012:**
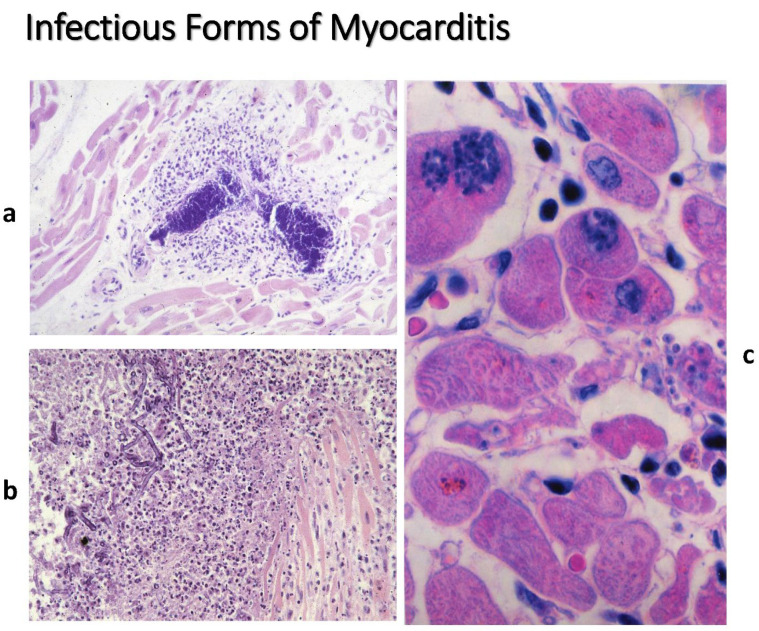
Infectious myocarditis. Note: interstitial Gram-positive bacteria (**a**), fungi (**b**), intracellular cytomegalovirus (**c**). Gram stain (**a**). Haematoxylin and Eosin stain (**b**,**c**).

**Figure 13 biomedicines-12-00832-f013:**
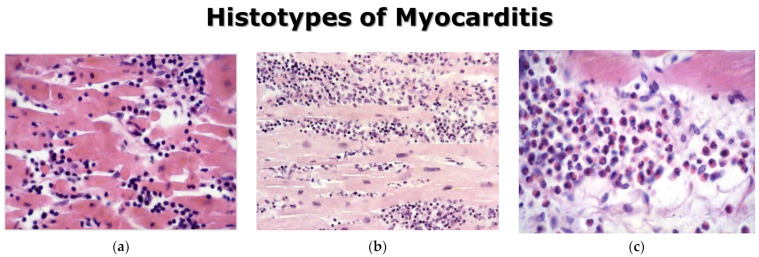
Histotypes of myocarditis: lymphocytic (**a**), neutrophil (**b**), eosinophil (**c**), granulomatous, non-caseous (**d**), giant cells (**e**). Haematoxylin and Eosin stain.

**Figure 14 biomedicines-12-00832-f014:**
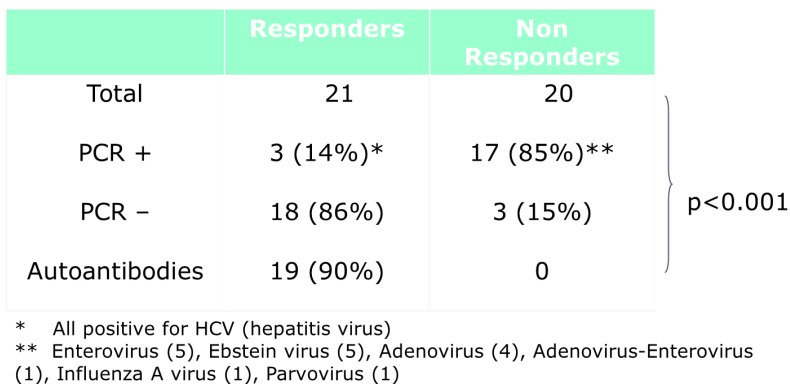
Immunosuppressive therapy is quite efficient for non–positive PCR and less for positive PCR myocarditis From [11], modified.

**Figure 15 biomedicines-12-00832-f015:**
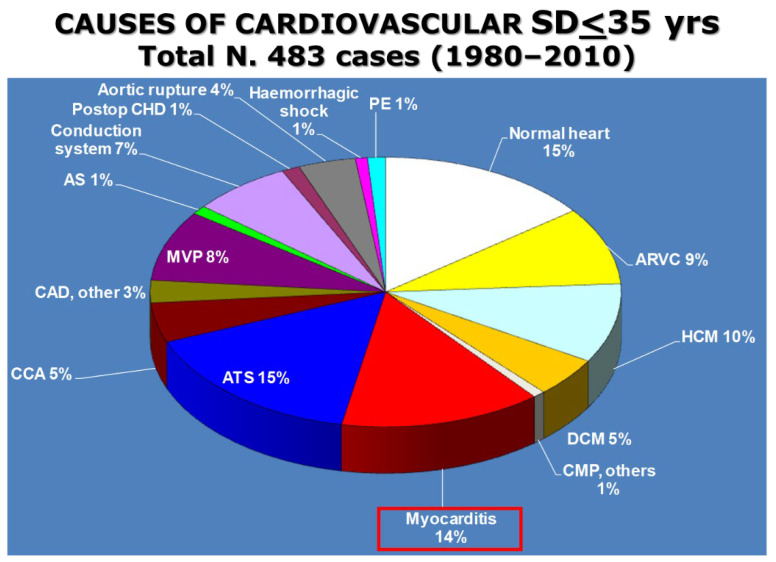
Findings from the Veneto Region Registry, Italy, 1980–2010: myocarditis accounts for 14% of juvenile SD (<35 years old). ARVC = arrhythmogenic right-ventricular cardiomyopathy; AS = aortic stenosis; ATS = coronary atherosclerotic disease; CAD = other coronary artery disease; CCA = coronary artery anomalies; CHD = congenital heart disease; CMP= other cardiomyopathies; DCM = dilated cardiomyopathies; HCM = hypertrophic cardiomyopathy; MVP = mitral valve prolapse; PE = pulmonary embolism; SD = sudden death. The red box was used to emphasize the role of myocarditis in juvenile SD.

**Figure 17 biomedicines-12-00832-f017:**
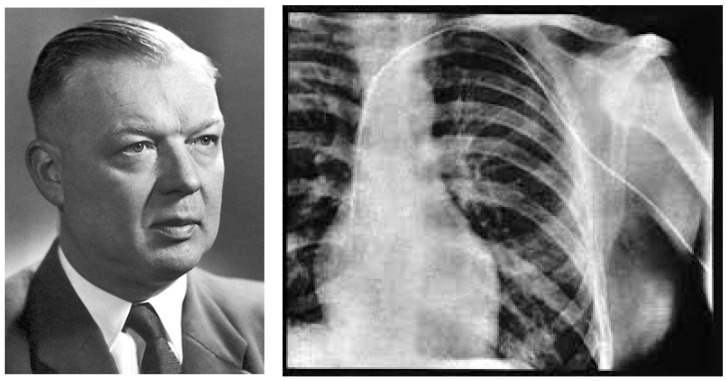
In 1929, Werner Forssmann (1904–1979) invented cardiac catheterism. He entered his own right ventricle with a urological catheter through the left radial vein.

**Figure 18 biomedicines-12-00832-f018:**
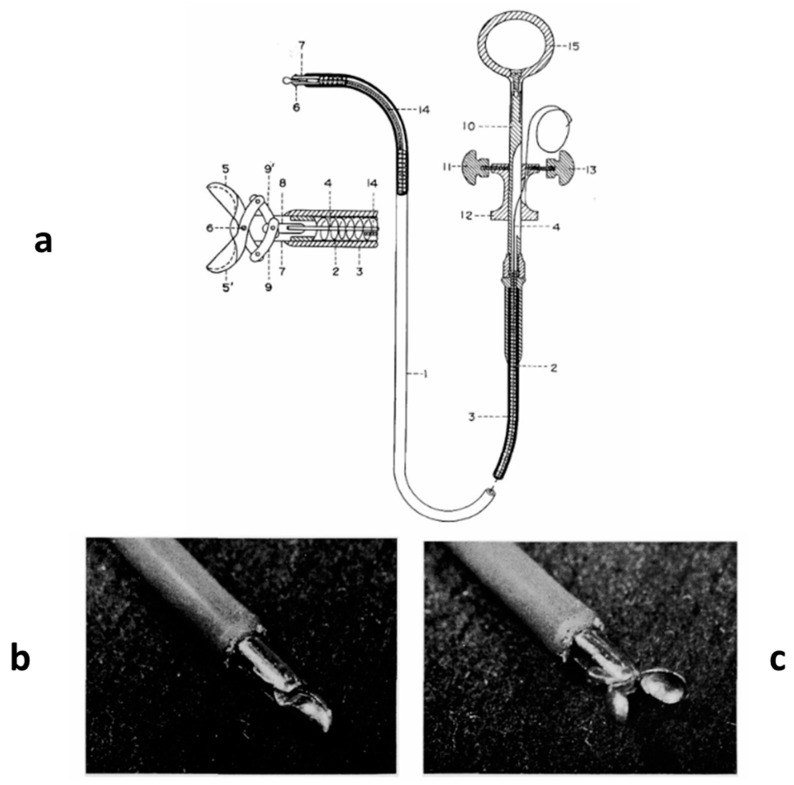
In 1962, Sakakibara (1910–1979) introduced in Japan the technique of transvenous endomyocardial biopsy (**a**). Note the bioptome at the extremity (**b**,**c**).

**Figure 19 biomedicines-12-00832-f019:**
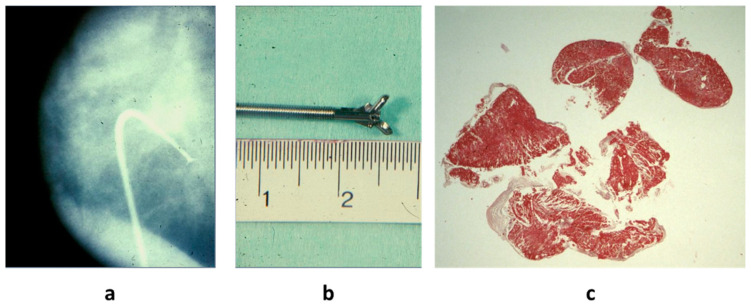
Current endomyocardial biopsy (Richardson catheter) along the inferior vena cava (**a**) by bioptome (**b**) with multiple samples (**c**). Haematoxylin and Eosin stain.

**Figure 20 biomedicines-12-00832-f020:**
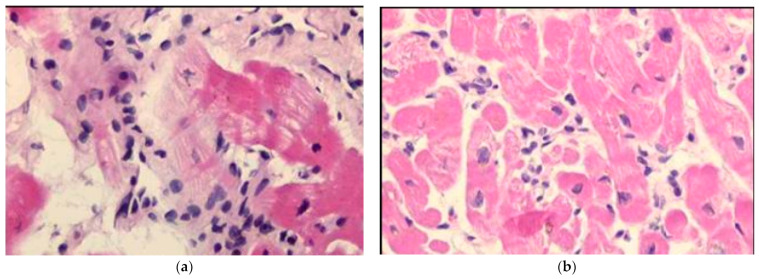
In 1985, Dallas Criteria were put forward for myocarditis diagnosis by endomyocardial biopsy. (**a**) Active myocarditis with myocardial necrosis; (**b**) Recurrent myocarditis. Haematoxylin and Eosin stain.

**Figure 21 biomedicines-12-00832-f021:**
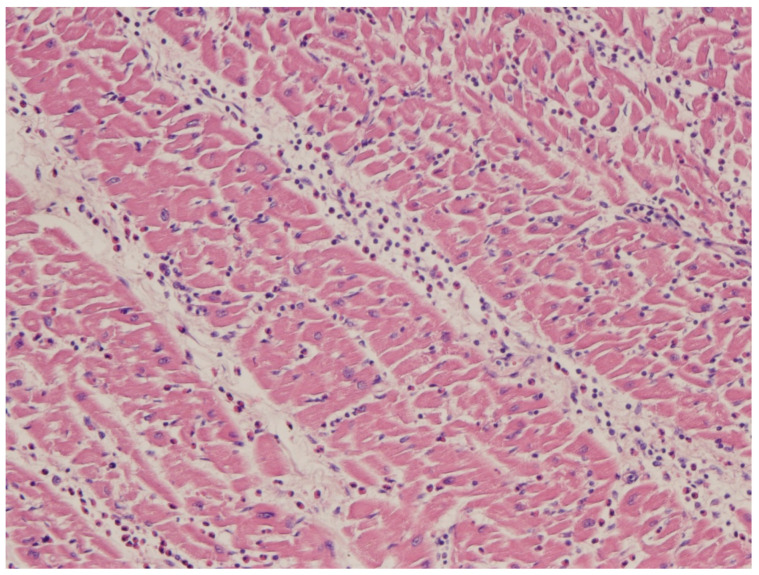
Myocarditis with inflammatory infiltrates and remarkable interstitial edema, in the absence of myocardial necrosis. Haematoxylin and Eosin stain.

**Figure 22 biomedicines-12-00832-f022:**
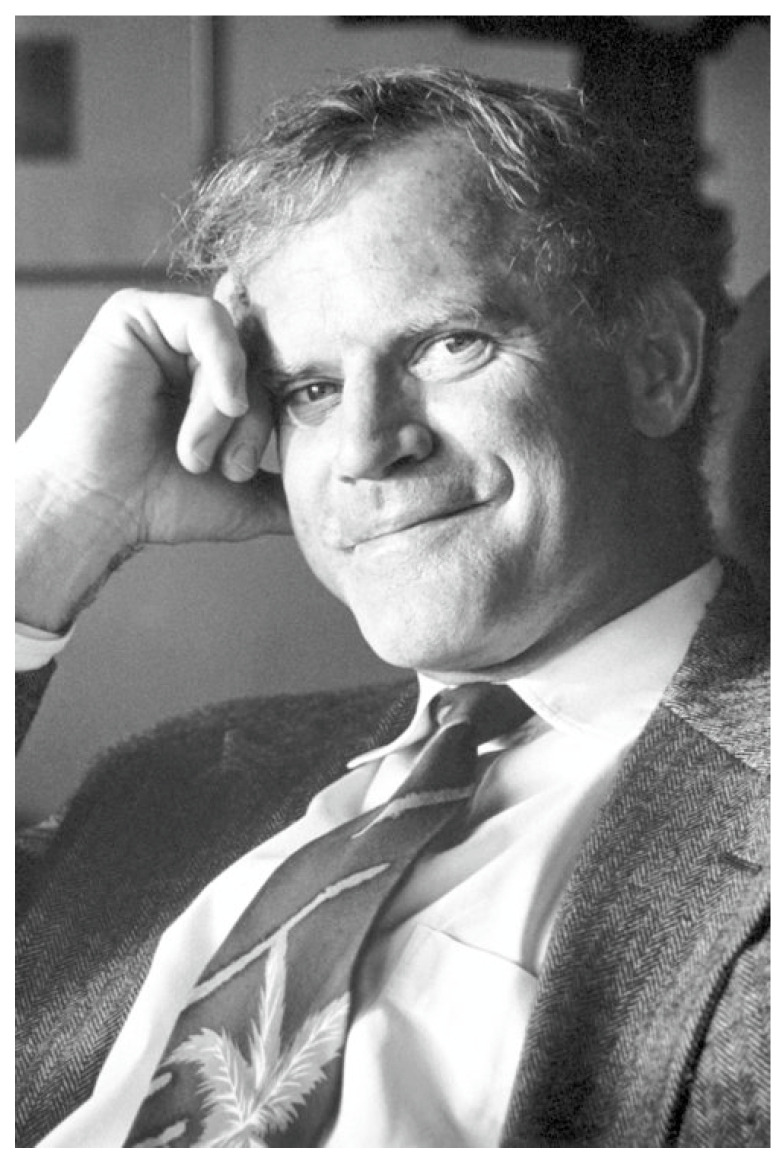
In 1986, Kary Mullis (1944–2019) invented Polymerase Chain Reaction (PCR) and received the Nobel Prize in 1993.

**Figure 23 biomedicines-12-00832-f023:**
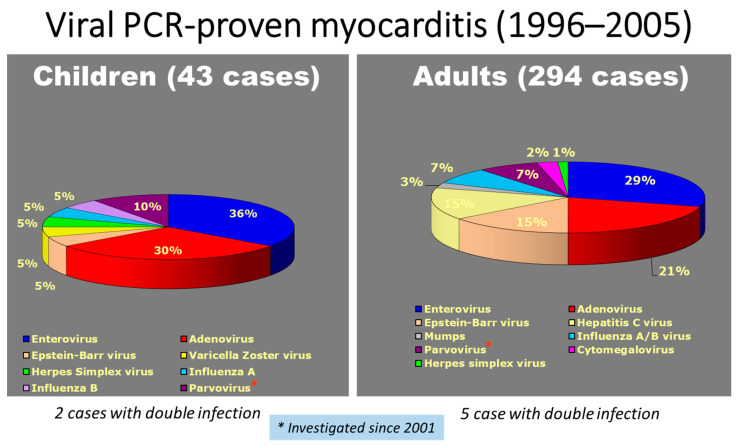
Incidence in children vs. adults of viral PCR-proven myocarditis (1996–2005, Cardiac Registry, Veneto Region, Italy).

**Figure 24 biomedicines-12-00832-f024:**
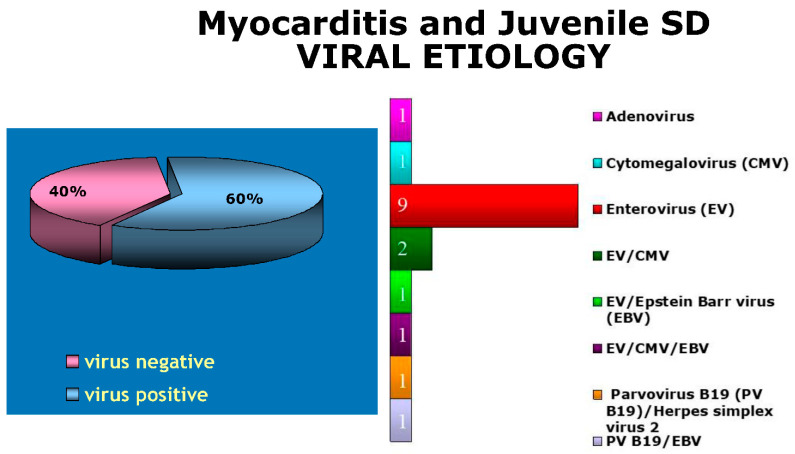
Enterovirus is the most frequent cause of sudden death by viral myocarditis (Veneto Region, Italy).

**Figure 25 biomedicines-12-00832-f025:**
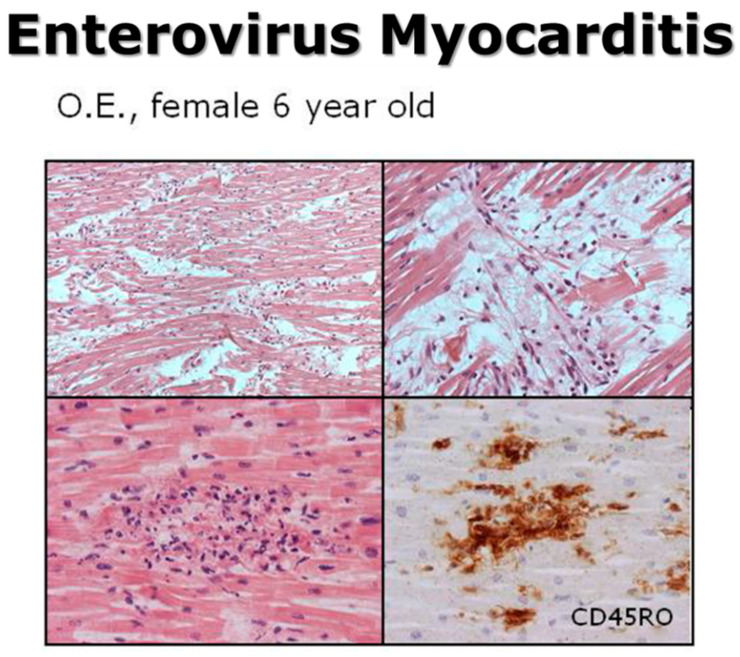
Sudden death by myocarditis with PCR detection of enterovirus (“molecular autopsy”). Haematoxylin and Eosin stain. CD45RO immunohistochemistry.

**Figure 26 biomedicines-12-00832-f026:**
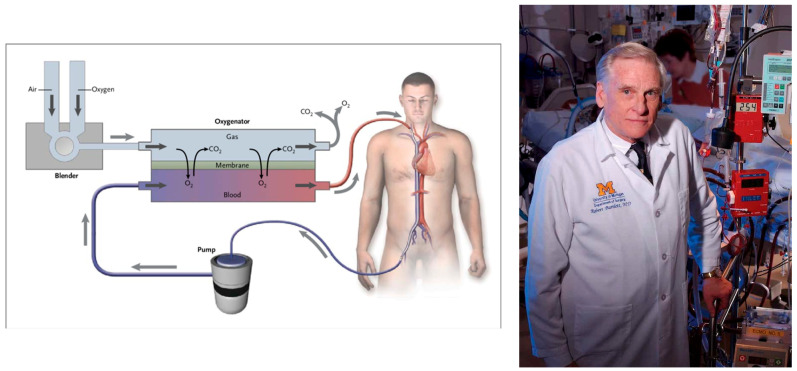
In 1970, Robert Bartlett (1939–) invented ExtraCorporeal Membrane Oxygenation (ECMO).

**Table 1 biomedicines-12-00832-t001:** Myocarditis—Infective Agents.

Viruses	DNA: Adenovirus, Herpes virus (cytomegalovirus, Epstein–Barr virus, varicella-zoster, Human Herpes Virus 6), Hepatitis B, Parvovirus B19
	RNA: Arbovirus, Hepatitis C virus, Picornavirus (entero, rhino), Orthomyxovirus (influenza A,B), Paramyxovirus (rubeola, mumps, respiratory syncytial virus), Retrovirus (HIV-1)
Bacteria	Staphylococcus, Streptococcus, Pneumococcus, Meningococcus, Gonococcus, Salmonella, Corynebacterium diphtheriae, Hemophilus influenzae, Mycoplasma pneumoniae, Brucella species
Mycobacteria	Tuberculosis, Avium intracellular, Leprae
Fungi	Aspergillus, Candida, Actinomyces, Blastomyces, Cryptococcus, Histoplasma
Protozoa	Toxoplasma gondii, Trypanosoma cruzi
Rickettsiae	Coxiella burnetti (Q fever), Rickettsia rickettsii, Rickettsia tsutsugamuschi
Chlamydia	Trachomatis, Pittaci
Parasitic	Trichinella spiralis

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
