# Peer review of "Storytelling of Myocarditis"

_biomedicines, 2024, doi:10.3390/biomedicines12040832_

Round 1

Reviewer 1 Report

Comments and Suggestions for Authors

It was really a great pleasure to read this “Storytelling of myocarditis” by Prof. Gaetano Thiene.  The most important facts and events that have led us to the moment when we can definitively determine whether we have myocarditis or not using a non-invasive method using a CMR examination.

The great efforts of scientists in identifying the causes of myocarditis have been very well highlighted. The most important milestones regarding treatment methods including ECMO are also shown.

We still cannot effectively prevent fulminant forms of myocarditis because we do not use antiviral vaccines effective, mostly against Enteroviruses, on  large scale.

This review of information highlights “the death of the Dallas criteria”, which means that the sine qua non of inflammation is only a diffuse inflammatory infiltrate without cardiomyocyte necrosis.

It may be worth mentioning that such conditions are common in autoimmune diseases, e.g. SLE and subsequent relapses of the disease do not lead to rapid declines in contractility.

In the abstract, the Author mentions the CAR Coxsackie-Adenovirus receptor, which is a common entry for these two viruses only.

There is no information that, other viruses, like Parvovirus B19 are vasculotropic, Herpes viruses  infect interstitial cells. The tropism of  Coronaviruses, influenza and hepatitis C virus would be also important to present.

 Some other very  minor remarks:

The information on the positioning of myocardits within WHO classifications of 1980 and 1995 is  in two places:

Page 12/26  lines 142-144

Page 21/26 lines 253-256

It looks like a repetition.

Some hiccups:

Page 21/26,  line 237: a food prognosis is feasible…  ? Maybe this could be reworded.

Page 21/26: line 248

“Application of ECMO allow temporary native heart relaxation… may be: “relieving the overloaded native heart”

Page 21/26 line 258: subitaneous onset

Page 19/26: line: 219

As for the prognosis, viral myocarditis shows less survival.   Maybe this could be reworded.

Spelling mistakes:

Page 19/26, line:  210: caridomiopathy

These minor remarks do not diminish the author's great contribution to building this field of knowledge based on pathomorphology, as well as in cooperation with outstanding clinicians from the University of Padua. This is a really much needed review.

Author Response

I thank a lot reviewer 1 for the flattering comments to my paper.

The following modifications have been introduced.

  • It may be worth mentioning that such conditions are common in autoimmune diseases, e.g. SLE and subsequent relapses of the disease do not lead to rapid declines in contractility.

I added that “such conditions are common in autoimmune diseases, e.g. SLE myocarditis and subsequent relapses of the disease do not lead to rapid decline in contractility, probably because the main target are the conducting tissues, with onset of sino-atrial or atrioventricular block”.

  • There is no information that, other viruses, like Parvovirus B19 are vasculotropic, Herpes viruses infect interstitial cells. The tropism of Coronaviruses, influenza and hepatitis C virus would be also important to present.

It has been added that Parvovirus B19 is mostly vasculotropic and herpes virus infects mesenchymal cells. Coronaviruses, influenza and hepatitis C virus appear in the Table I among cardiotropic viruses.

  • Minor remarks.

I added:

  • The position of myocarditis within WHO classification appears in all the subsequent classifications.
  • One of the repetitions have been removed
  • Hiccups: “a food prognosis is feasible…” Sorry I meant good. The statement has been reworked as follow: a good prognosis is feasible.
  • In the abstract I changed “Application of ECMO allows temporary native heart relaxation…” with: “Application of Extracorporeal Membrane Oxygenation (ECMO) allows relieving the overloaded native heart”.
  • Page 21/26 line 258: subitaneous onset. It was reworked as follow: “As for the outcome, viral myocarditis shows the worst prognosis”.
  • The spelling mistake of cardiomiopathy has been corrected with cardiomyopathy.

Reviewer 2 Report

Comments and Suggestions for Authors

The review article 'Storytelling of myocarditis' written by Gaetano Thiene is interesting but needs several changes before it should be published.

1.) The iThenticate report identifies several paragraphs where the author has reused sentences from his own previously published manuscript. The author should rewrite these paragraphs as indicated by the iTheniticate reports.

2.) Not all abbreviations like PCR are explained, when they were used the first time.

3.) The English should be corrected by a native speaking editor. Example: Cardiac magnetic resonsonance (line 18, abstract). Several other mistakes are present in the text.

4.) The source of the images is unclear. Please add relevant references for each image.

5.) In histology images the scale bars are missing.

6.) I would also discuss cardiac inflamation caused by genetic defects / mutations. For example the DSC2 transgenic mouse (Brodehl A et al. 2019 PLOSone) develops severe cardiomyopathy caused by inflammation. I would mention this, that cardiac inflammation can be even caused by genetic factors und would discuss this finding.

7.) Figure 15: The Size Marker should be labbeled in this image!

8.) Figure 24: The size marker should be also labelled.

In summary, I suggest a major revision for this manuscript.

Author Response

I thank Reviewer 2 very much indeed for the constructive comments to revise the manuscript.

  • The iThenticate report identifies several paragraphs where the author has reused sentences from his own previously published manuscript. The author should rewrite these paragraphs as indicated by the iTheniticate reports.

The above mentioned paragraphs have been partly rewritten.

  • Not all abbreviations like PCR are explained, when they were used the first time.

PCR has been written per extenso when first used.

  • The English should be corrected by a native speaking editor. Example: Cardiac magnetic resonsonance (line 18, abstract). Several other mistakes are present in the text.

The English has been checked by a native speaker, correcting the several mistakes.

  • The source of the images is unclear. Please add relevant references for each image.

The majority of the histologic pictures is original.

  • In histology images the scale bars are missing.

Our custom is to use scale bars only for ultrastructural pictures. We prefer to specify the staining used.

  • I would also discuss cardiac inflammation caused by genetic defects / mutations. For example the DSC2 transgenic mouse (Brodehl A et al. 2019 PLOSone) develops severe cardiomyopathy caused by inflammation. I would mention this, that cardiac inflammation can be even caused by genetic factors and would discuss this finding.

Myocarditis was one of the mechanisms postulated in the pathophysiology of AC since the origin (C Basso, G Thiene, D Corrado, A Angelini, A Nava, M Valente. Arrhythmogenic right ventricular cardiomyopathy. Dysplasia, dystrophy, or myocarditis? Circulation. 1996 Sep 1;94(5):983-91. doi: 10.1161/01.cir.94.5.983. PMID: 8790036.).

Virus negative, probably autoimmune myocarditis was found at EMB in the so called “hot phase of arrhythmogenic cardiomyopathy (AC), especially in children (R Bariani, A Cipriani, S Rizzo, R Celeghin, M Bueno Marinas, B Giorgi, M De Gaspari, I Rigato, L Leoni, A Zorzi, M De Lazzari, A Rampazzo, S Iliceto, G Thiene, D Corrado, K Pilichou, C Basso, M Perazzolo Marra, B Bauce. 'Hot phase' clinical presentation in arrhythmogenic cardiomyopathy. Europace. 2021 Jun 7;23(6):907-917. doi: 10.1093/europace/euaa343. PMID: 33313835; PMCID: PMC8184227.).

Transgenic mice overexpressing desmocollin-2 was found to develop AC associated with myocardial inflammation (A Brodehl, DD Belke, L Garnett, K Martens, N Abdelfatah, M Rodriguez, C Diao, YX Chen, PM Gordon, A Nygren, B Gerull. Transgenic mice overexpressing desmocollin-2 (DSC2) develop cardiomyopathy associated with myocardial inflammation and fibrotic remodeling. PLoS One. 2017 Mar 24;12(3):e0174019. doi: 10.1371/journal.pone.0174019. PMID: 28339476; PMCID: PMC5365111.).

These papers have been quoted among references.

  • Figure 15: The Size Marker should be labbeled in this image!

Figure 15 has been deleted.

  • Figure 24: The size marker should be also labelled.

In the legend I added the staining used, as did for the other histologic pictures.

Round 2

Reviewer 2 Report

Comments and Suggestions for Authors

The author has improved his manuscript. I suggest to accept it for publication.